# An In-depth Study of Stochastic Backpropagation

**Jun Fang**    **Mingze Xu**[*]   **Hao Chen**   **Bing Shuai**   **Zhuowen Tu**   **Joseph Tighe**

AWS AI Labs

{junfa, xumingze, hxen, bshuai, ztu, tighej}@amazon.com

## Abstract

In this paper, we provide an in-depth study of Stochastic Backpropagation (SBP) when training deep neural networks for standard image classification and object detection tasks. During backward propagation, SBP calculates gradients by using only a subset of feature maps to save GPU memory and computational cost. We interpret SBP as an efficient way to implement stochastic gradient decent by performing backpropagation dropout, which leads to significant memory saving and training run-time reduction, with a minimal impact on the overall model accuracy. We offer best practices to apply SBP for training image recognition models, which can be adopted in learning a wide range of deep neural networks. Experiments on image classification and object detection show that SBP can save up to 40% of GPU memory with less than 1% accuracy degradation. Code is available at: https://github.com/amazon-research/stochastic-backpropagation

## 1 Introduction

A common practice to improve accuracy when training deep neural networks is to increase the input resolution [19], model depth [7], and model width [21]. However, this can significantly increase the GPU memory usage, making training on devices with limited resources difficult. For example, training an object detector using ConvNeXt-Base [14] as the backbone with batch size 2 requires over 17 GB of memory, which cannot fit on many modern GPUs.

A number of prior works have explored memory-efficient training designs, especially for large model training on high resolution images or videos. Mixed-precision training [17] uses lower precision to represent the network weights and activations for certain layers of the network. Gradient checkpointing [2] restores intermediate feature maps by recomputing nodes in the computation graph during the backward pass, which requires extra computations. Gradient accumulation [12] splits a mini-batch into smaller chunks to calculate gradients iteratively, which slows down training process.

A recent method, Stochastic Backpropagation (SBP) [3], offers a new perspective for memory-efficient model training. It performs partial execution of gradients during backpropagation on randomly selected frames when training video models [13, 24, 23, 22]. It achieves considerable memory saving by only caching a subset of video frame feature maps without significant accuracy loss. However, SBP in [3] is only applied to video-based models using two architectures and its design space has not be thoroughly explored on wider tasks. The explanation provided in [3] for the effectiveness of SBP is simple but narrow — it attributes the validity of the method to the high redundancy of video frames.

This leaves a number of questions around SBP unanswered: (1) Is SBP 's effectiveness limited to the temporal redundancy of video or can it effectively leverage spatial redundancy? (2) If so, can SBP be generalized to image-based tasks, with more limited redundancy? (3) Is SBP generalizable

---

[*]Corresponding Author.

to different networks (*e.g.,* MLP, CNN, Transformer) and operators? (4) What are the key design choices when implementing SBP ?

To answer these questions, we first formalize the SBP process and analyze the effect of SBP on the calculation of gradients. With this formulation, we show how to generalize the idea of SBPto image models and present an in-depth study of the design choices when using SBP. During backward propagation, SBP calculates gradients from a subset of spatial feature maps to save memory and computation cost. We observe that the calculated gradients by SBP are highly correlated with the standard SGD gradients, indicating that SBP is a reasonable approximation. We also provide a new, straightforward implementation of the idea that requires only a few lines of code. We explore several important design strategies, which have a noticeable impact on the performance. We validate the generalizability of SBP on two common tasks (image classification and object detection) with two popular network architectures (ViT [6] and ConvNeXt [14]). We show that SBP can effectively train image models with less than 1% accuracy loss in a memory-efficient fashion.

## 2 Related Work

Since deep neural networks have expanded in both depth [7] and breadth [21] in recent years, the area of designing memory-efficient training methods has attracted a fair amount of attention. By saving and recomputing nodes in the computation graph during backpropagation, gradient checkpointing [2] can save a large amount of memory. Gradient accumulation [12] splits the large sample batch into several mini batches and runs them iteratively without updating model parameters. This saves the GPU memory proportionally without affecting the accuracy, because accumulating the gradients of these sub-iterations and then updating the model parameters are identical to directly optimizing the model with a global batch size. However, these two methods slow down the training process. Multigrid [20] proposes to use different mini-batch shapes to speedup the model training. Sparse network [5] particularly targets the recognition tasks but can only save the memory theoretically. Sideways [15] and its follow-up [16] reduce the memory cost by overwriting activations whenever new ones become available, but they are limited to causal models. SBP [3] reduces a large portion of GPU memory by only backpropagating gradients from incomplete execution for video models. In this paper, we generalize SBP to more standard computer vision tasks including image recognition.

## 3 Understanding Stochastic Backpropagation (SBP)

In this section, we first formalize the process of Stochastic Backpropagation (SBP) (sec. 3.1) and show how to compute the gradient for two common operations (sec. 3.2). With these formulations established, we analyze how to perform the chain rule of gradient calculations for SBP (sec. 3.3). Finally, we present a simple implementation of SBP in practice with only a few lines of code (sec. 3.4).

### 3.1 Formulation of SBP

We first present a general formulation of SBP [3]. We denote a loss function $L(\Theta)$ to learn a model on a dataset $D = \{d_j\}_{j=1}^N$, where $\Theta$ is the collection (vector, matrix, or tensor) of all parameters of the model, $N$ is the total number of training samples, $d_j \in \mathbb{R}^{T \times H \times W \times C}$ is an input data sample, where $H$ and $W$ are spatial resolution height and width respectively, $C$ is the input channel size, $T = 1$ represents an image and $T > 1$ represents a video.

In each training iteration, the model with SBP processes a mini-batch of data samples $S \in \mathbb{R}^{B \times T \times H \times W \times C} \subseteq D$ to compute the loss after the forward pass, where $B$ is the batch size. During the backward pass, different from the traditional backpropagation that relies on the full feature maps $X_i \in \mathbb{R}^{B \times T_i \times H_i \times W_i \times C_i}$($i$ is the index of the layers), an incomplete execution for backpropagation (*i.e.,* SBP) only utilizes a subset of the feature maps $X_i^{sub} \in \mathbb{R}^{B \times T_i^{sub} \times H_i^{sub} \times W_i^{sub} \times C_i}$ to approximate the gradient calculation. Specifically, it updates the model weights as

$$\Theta_{iter+1} = \Theta_{iter} - \eta \frac{1}{B} \sum_{x_i^{sub} \in X_i^{sub}} \nabla_\Theta L(x_i^{sub}, \Theta_{iter}), \tag{1}$$

where $\eta$ is the learning rate, $\nabla_\Theta$ is the collection of model weight gradients, $\Theta_{iter}$ and $\Theta_{iter+1}$ are model weights at the current and next iteration, respectively.

SBP computes all forward paths, but only caches a subset of feature maps in the backward pass. Note that the method in [3] is a special case of (1) as it operates on the temporal dimension by using a subset of video frames $X_i^{T_i^{sub}} \in \mathbb{R}^{B \times T_i^{sub} \times H_i \times W_i \times C_i}$.

## 3.2 Backward Gradient Calculation

Next, we derive equations to compute the gradient during the backward phase of SBP. We denote that at layer $i$ of the model, an operator $f_i$ with learnable weights $W_{f_i}$ to processes the feature maps $X_i \in \mathbb{R}^{B \times T_i \times H_i \times W_i \times C_i^{in}}$, where $C_i^{in}$ refers to the input channel size.

### 3.2.1 Calculation of Linear Layers

We start by exploring $f_i$ as a point-wise convolutional (PW-Conv) layer. PW-Conv is widely used in the state-of-the-art computer vision models such as ViT [6] and ConvNeXt [14]. It is equivalent to the linear layers in Transformer MLP blocks [14], and is also known as channel-wise fully connected layer. It can be either implemented through the linear layer, or the $1 \times 1$ convolutional layer. It is mainly used to enrich the channel information with weights $W_{f_i} \in \mathbb{R}^{C_i^{in} \times C_i^{out}}$, and the analysis on PW-Conv can be easily extended to other layers.

The forward pass of this layer is (the bias term is omitted for simplicity):

$$X_{i+1} = f_i(X_i) = X_i W_{f_i}, \tag{2}$$

where $X_{i+1} \in \mathbb{R}^{(B \times T_i \times H_i \times W_i) \times C_i^{out}}$ ($X_i$ and $X_{i+1}$ are reshaped as a matrix format). During the traditional backward pass with full backpropagation, the gradients of both activations and weights are calculated by:

$$\begin{aligned}
dX_{i+1} &= \partial Loss/\partial X_{i+1}, \\
dW_{f_i} &= \partial Loss/\partial W_{f_i} = X_i^T dX_{i+1}, \\
dX_i &= \partial Loss/\partial X_i = dX_{i+1} W_{f_i}^T.
\end{aligned} \tag{3}$$

It requires storing the weights $W_{f_i} \in \mathbb{R}^{C_i^{in} \times C_i^{out}}$ and full feature maps $X_i \in \mathbb{R}^{(B \times T_i \times H_i \times W_i) \times C_i^{in}}$ to compute the gradients. In many cases, $B \times T_i \times H_i \times W_i \gg C_i^{out}$, i.e., caching of feature maps dominates the memory usage. In order to effectively save GPU memory during training, SBP only caches a subset of the feature maps for backpropagation.

In the backward pass of SBP process, we split the full feature maps $X_i$ into two subsets $X_i^{keep}$ and $X_i^{drop}$, which denote the corresponding indices of the sampled feature maps that are *kept* and *dropped*, respectively. An example of $X_i^{keep}$ can be $X_i[:, \text{even index on } T, :, :, :]$ for video input, or $X_i[:, :, \text{even index on } H, \text{odd index on } W, :]$ for image input (e.g., Fig. 3 left), and $X_i^{drop}$ is the complementary part of the feature maps. We have $X_i^{keep} \cap X_i^{drop} = \emptyset$ and $X_i^{keep} \cup X_i^{drop} = X_i$. We denote keep-ratio $r$ as the number of gradient *kept* indices $j \in \mathbb{Z}_i^{keep}$ over the number of all indices.

One advantage of simplifying the SBP problem by using PW-Conv operator is that the nodes in spatial or temporal dimensions can be calculated independently through the forward and backward passes. Therefore, we can update the forward pass separately on the *kept* and *dropped* indices:

$$\begin{aligned}
X_{i+1} &= [X_{i+1}^{keep}, X_{i+1}^{drop}] = [X_i^{keep}, X_i^{drop}] W_{f_i}, \\
\text{i.e.,} \quad X_{i+1}^{keep} &= X_i^{keep} W_{f_i}, \quad X_{i+1}^{drop} = X_i^{drop} W_{f_i}
\end{aligned} \tag{4}$$

as well as the backward pass:

$$\begin{aligned}
dX_{i+1} &= [dX_{i+1}^{keep}, dX_{i+1}^{drop}], \\
dX_i^{keep} &= dX_{i+1}^{keep} W_{f_i}^T, \quad dX_i^{drop} = dX_{i+1}^{drop} W_{f_i}^T, \\
dW_{f_i} &= X_i^T dX_{i+1} = [X_i^{keep}, X_i^{drop}]^T [dX_{i+1}^{keep}, dX_{i+1}^{drop}] \\
&= X_i^{keep^T} dX_{i+1}^{keep} + X_i^{drop^T} dX_{i+1}^{drop}.
\end{aligned} \tag{5}$$

SBP drops all the gradients calculation with superscripts $^{drop}$ and keeps all the gradients calculation with superscripts $^{keep}$, thus it saves memory and computation on the dependencies of *dropped* indices. In other words, the memory usage and the backward computation cost are proportional to the keep-ratio of the original case. Mathematically, it is equivalent to set $dX_{i+1}^{drop} = \mathbf{0}$ (thus $dX_i^{drop} = \mathbf{0}$) and

uses $dW_{f_i}^{keep} = X_i^{keep^T} dX_{i+1}^{keep}$ as a stochastic approximation of the original weight gradients $dW_{f_i}$ to update the model weights.

### 3.2.2 Calculation of Convolutional Layers

We then derive the calculation of a general convolutional layer (ConvLayer) as $f_i$. Unlike the PW-Conv, nodes of general ConvLayer with kernel size $> 1$ in spatial or temporal dimensions are no longer independent in forward and backward calculations. Assume that we apply a convolutional kernel $k \times k$ with stride $s$. The regular backpropagation of the ConvLayer is calculated as follows:

$$dX_i = zeropad(dX_{i+1}) * W'_{f_i}$$
$$dW_{f_i} = X_i * dX_{i+1} \tag{6}$$

where $*$ is the convolution operation, $zeropad$ is to pad the edge of the matrix with zeros, and $W'_{f_i}$ represents the $180°$ rotation of $W_{f_i}$.

From Eq. (6), in the gradient calculation of ConvLayer at a given location $j$ on the input feature map as $x_i^j \in X_i$, all the elements from $dX_{i+1}$ inside the region of the kernel size $k$ will contribute to the gradient calculation of $dx_i^j$. Therefore, in SBP of ConvLayer, when we set $dX_{i+1}^{drop} = \mathbf{0}$ for certain areas, in general, two things will happen: (a) even if $j \in \mathbb{Z}^{drop}$ and $dx_{i+1}^j = 0$, $dx_i^j$ will not be 0 due to the non-zero gradient contribution of the neighbors of $j$ on $dX_{i+1}$; (b) $dx_i^j$ will be an approximated gradient unless all $dx_{i+1}^p$ with $p$ in the neighbor areas of $j$ in terms of the kernel size $k$ (*i.e.,* indices $p$ are within the convolutional kernel of index $j$) are kept and there is no chain rule effect on $dx_{i+1}^p$. However, there is a special scenario: when $s \geq k$, the neighbor effects disappear due to the fact that $dX_{i+1}$ needs to be zero interweaved in gradient calculation. As a result, nodes in spatial or temporal dimensions are patch-wise independent, which is similar to the case of PW-Conv.

### 3.2.3 Discussion

From the above derivation, we see that the approximation effect of SBP performs differently when the operator changes. For layers, such as linear layer or PW-Conv, the activation gradients (*i.e., $dx_i$*) are either $\mathbf{0}$ or exact[2]. While for convolutional layers with kernel size larger than 1, most of the gradients on the activations are neither $\mathbf{0}$ nor exact. Due to space limitation, we only derived SBP on two most popular operators. For derivation of other operators, please refer to the supplementary material.

### 3.3 Chain Rule Effect

We have derived the calculation of SBP on a single layer in the previous section. In this section, we derive the calculation of SBP on a stack of multiple layers. For simplicity, we take two layers as an example, which can be easily generalized to more layers.

### 3.3.1 Two PW-Conv Layers

We first consider a scenario with two PW-Conv layers. We assume a layer before $f_i$, *i.e., $f_{i-1}$*, is another PW-Conv layer with weights $W_{f_{i-1}}$. The forward pass of this layer is (the bias term is also omitted):

$$X_i = f_{i-1}(X_{i-1}) = X_{i-1}W_{f_{i-1}}. \tag{7}$$

Then we can calculate the gradients of weights and activations in layer $f_{i-1}$ similar to Eq. (3):

$$dX_{i-1} = dX_i W_{f_{i-1}}^T = dX_{i+1} W_{f_i}^T W_{f_{i-1}}^T,$$
$$dW_{f_{i-1}} = X_{i-1}^T dX_i = X_{i-1}^T dX_{i+1} W_{f_i}^T. \tag{8}$$

We first consider the case when there is no *additional* gradient dropout at layer $f_{i-1}$. Since $dX_{i+1}$ has non-zero gradients at *kept* indices (*i.e., $dX_{i+1}^{keep}$*) and zero gradients at *dropped* indices (*i.e., $dX_{i+1}^{drop}$*), Eq. (8) can be updated to:

$$dX_{i-1} = [dX_{i-1}^{keep}, dX_{i-1}^{drop}],$$
$$dX_{i-1}^{keep} = dX_{i+1}^{keep} W_{f_i}^T W_{f_{i-1}}^T, \quad dX_{i-1}^{drop} = dX_{i+1}^{drop} W_{f_i}^T W_{f_{i-1}}^T = \mathbf{0}, \tag{9}$$
$$dW_{f_{i-1}} = X_{i-1}^{keep^T} dX_{i+1}^{keep} W_{f_i}^T.$$

---

[2]By "exact" we mean the exact calculation of the gradient for the given mini-batch, though it is still an estimation due to SGD.

From the equation, we can see that if there is no *additional* gradient dropout applied on layer $f_{i-1}$, through the chain rule, the *keep* and *drop* set will transfer from $X_{i+1}$ to $X_{i-1}$ identically.

If we apply gradient dropout on layer $f_{i-1}$ as well, *i.e.*, on layer $f_{i-1}$, we split $dX_i$ into $dX_i^{keep}$ and $dX_i^{drop}$. Assume that the *keep* and *drop* indices set at layer $i+1$ is $\mathbb{Z}_{i+1}^{keep}$ and $\mathbb{Z}_{i+1}^{drop}$, and at layer $i$ is $\mathbb{Z}_i^{keep}$ and $\mathbb{Z}_i^{drop}$, respectively. Then through Eq. (8) and Eq. (9), the *keep* and *drop* indices set on layer $i-1$ will be:

$$\mathbb{Z}_{i-1}^{keep} = \mathbb{Z}_{i+1}^{keep} \cap \mathbb{Z}_i^{keep}, \quad \mathbb{Z}_{i-1}^{drop} = \mathbb{Z}_{i+1}^{drop} \cup \mathbb{Z}_i^{drop}. \tag{10}$$

Eq. (9) becomes:

$$dX_{i-1} = [dX_{i-1}^{keep'}, dX_{i-1}^{drop'}],$$
$$dX_{i-1}^{keep'} = dX_{i+1}^{keep'} W_{f_i}^T W_{f_{i-1}}^T, \quad dX_{i-1}^{drop'} = \mathbf{0}, \tag{11}$$
$$dW_{f_{i-1}} = X_{i-1}^{keep'}{}^T dX_{i+1}^{keep'} W_{f_i}^T.$$

where $keep' = \mathbb{Z}_{i-1}^{keep}$ and $drop' = \mathbb{Z}_{i-1}^{drop}$.

### 3.3.2 PW-Conv + ConvLayers

Now let's look at the chain-rule effect of the case when $f_{i-1}$ is a ConvLayer with kernel size $k > 1$ and stride $s$. Using the chain rule, we can update the Eq. (6) as follows:

$$dX_{i-1} = zeropad(dX_i) * W'_{f_{i-1}} = zeropad(dX_{i+1} W_{f_i}^T) * W'_{f_{i-1}},$$
$$dW_{f_{i-1}} = X_{i-1} * dX_i = X_{i-1} * dX_{i+1} W_{f_i}^T. \tag{12}$$

We can see from above equation that each $dx_{i-1}^j$ receives the backward gradients from the neighbors of $dx_i^j$, which might be partially dropped out, thus the gradient at $dx_{i-1}^j$ is no longer exact.

From Eq. (12), the dropout at layer $i$ makes the matrix $dX_i$ sparse (only indices at $\mathbb{Z}_{i+1}^{keep}$ are non-zero). If we conduct dropout at the current layer $f_{i-1}$ (*i.e.*, we have $\mathbb{Z}_i^{keep}$ and $\mathbb{Z}_i^{drop}$), the matrix $dX_{i-1}$ will become even more sparse unless the dropout is happening on the same location $\mathbb{Z}_{i+1}^{drop} = \mathbb{Z}_i^{drop}$. If we stack PW-Conv + ConvLayer further and further in an interweaving way (*i.e.*, $\mathbb{Z}_{i+1}^{drop} \neq \mathbb{Z}_i^{drop}$), the chain rule effect may lead to a very sparse output derivative matrix (*e.g.*, $dX_{i-1}$). However, stacking ConvLayers (due to the space limitation, we skip the math derivation) will not have such a trend, as the gradient dropout in ConvLayer only creates approximated gradient but not many zeros, because of the neighbor effect of gradient calculation.

### 3.3.3 Discussion

Overall, in terms of the chain rule effect, we can see from Eq. (11) that stacking PW-Conv layers, especially in an interweaving way, may lead to a sparse keep set $\mathbb{Z}_{i-1}^{keep}$ and a vanishing gradient of $dW_{f_{i-1}}$. Thus, in practice, we recommend to drop gradients at the same positions across layers to avoid the gradient vanishing and maintain high accuracy. Stacking PW-Conv + ConvLayers (in Eq. (12)) also tends to make the keep set sparse and thus should be avoided. While stacking ConvLayers doesn't have such effect, it stacks the non-exact gradient at almost every position but tends to make the gradients gradually drift from the dense gradients.

### 3.4 Efficient Implementation

With a deeper understanding of how SBP works, we provide a simple and efficient implementation of the SBP technique in Alg. 1. We point out that this is more efficient than the prior work [3] as there is (1) no need to do the re-compute the forward during the backward pass, thus it is computationally more efficient, and (2) no need to manually cache the intermediate feature maps and fill zero gradients in the backward pass, which is much simpler. We emphasize that it is very simple and easy to implement, it only needs 3 lines of code to do the forward, and the backward (no need to implement manually) can be seamlessly handled by the ease of "autograd engine" in the deep learning frameworks. It supports point-wise convolutional layers, the dot-product attention layers, and multilayer perception (MLP) in popular networks including ViT [6] and its variants [6, 11, 13], ConvNeXt [14], and hybrid architectures [4].

**Algorithm 1** Pytorch-like pseudocode of SBP for an arbitrary operation $f$.

```
# f: an arbitrary operation
# grad_keep_idx: sampled indices where gradients are kept
# grad_drop_idx: sampled indices where gradients are dropped

def sbp_f(f, inputs, grad_keep_idx, grad_drop_idx):
    # initiate outputs
    outputs = torch.zeros(output_shape, device=inputs.device)
    # forward with gradient calculation, gradients will be calculated with torch.autograd
    with torch.enable_grad():
        outputs[grad_keep_idx] = f(inputs[grad_keep_idx])
    # forward without gradient calculation
    with torch.no_grad():
        outputs[grad_drop_idx] = f(inputs[grad_drop_idx])
    return outputs
```

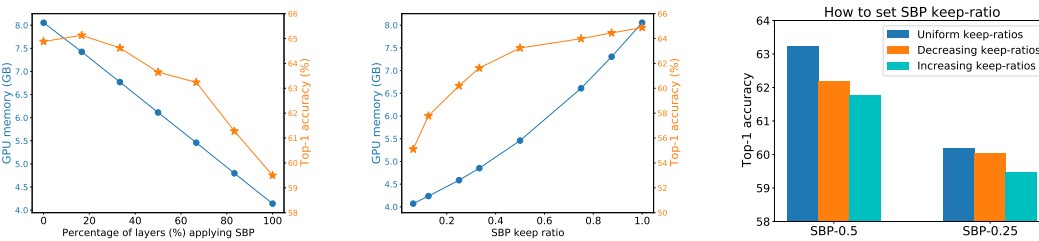

Figure 1: Illustration of the effect on applying SBP on different network layers and keep-ratios. Left: accuracy curve of applying SBP on different percentage of layers, the keep ratio of SBP is 0.5. Middle: trade-off of keep-ratio versus accuracy, every SBP layer has the same keep-ratio. Right: comparison of using uniform, decreasing and increasing keep-ratios for the SBP layers.

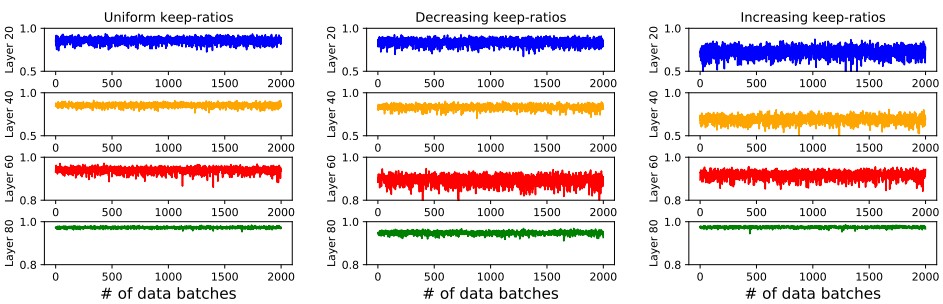

Figure 2: Cosine similarity of gradient weights with and without applying SBP on ViT-Tiny with uniform, decreasing, and increasing keep-ratios. Results on Layer 20 (MHSA), 40 (MLP), 60 (LayerNorm), 80 (MLP) are displayed. Results on other layers have similar behavior.

## 4 Design Strategies

In this section, we investigate the effectiveness of different design strategies of SBP on image tasks. We use ViT-Tiny [6] model to evaluate the top-1 accuracy on ImageNet [18] validation dataset. The ViT-Tiny is a smaller version of the ViT [6] variants, it uses patch size 16 with 12 transformer blocks (more than 80 layers), the embedding dimension is 192 and the number of heads is 3, i.e., each head has dimension 64. The experiments are trained for 100 epochs on the ImageNet dataset. The conclusions remain consistent for larger models and longer training schedules.

**How many layers should be applied with SBP?** To answer this question, we progressively apply SBP to transformer layers from the last layers to early layers and present their results in Fig. 1 Left. As shown, we observe a sweet spot when applying SBP to 8 out of 12 block layers, which reduces GPU memory consumption by 33% with an accuracy drop within 1.5%. Although applying SBP on all the transformer layers can save the more GPU memory, it causes a large accuracy drop. Therefore, unless stated otherwise, we apply SBP on 2/3 of all layers of ViT models for the rest of this work.

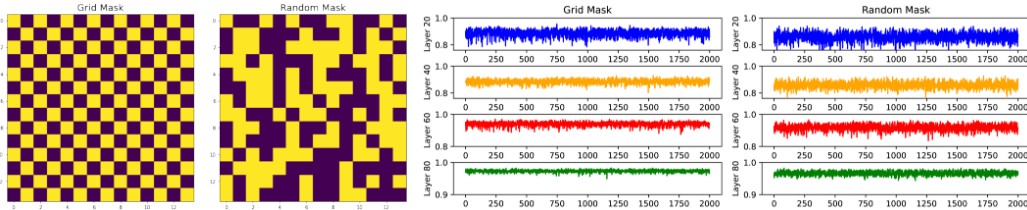

Figure 3: Left two: a particular example of gradient keep mask for grid-wise sampling and random sampling. Right two: cosine similarity of weights gradient between with and without applying SBP on ViT-Tiny. The SBP are applied with a keep-ratio of 0.5 on these two mask sampling methods.

**How much gradients should be kept in SBP?** In this section, we discuss the trade-off between the gradient keep-ratio and accuracy in SBP. As a reminder, keep-ratio $r$ is the number of gradient kept indices over the number of all indices, and it represents the percentage of spatial feature map values used to calculate the gradients. In general, the higher the keep-ratio is, the more gradient information is preserved, and the higher the accuracy is, the less memory is saved. A natural question arises: what is a good keep-ratio to balance the accuracy drop and memory saving?

In Fig. 1 Middle, we show the trade-off between model accuracy and the keep-ratio of gradients. In this case, we apply the same keep-ratio to all the SBP layers at the same gradient kept indices. We name this strategy "uniform keep-ratio". We observe that setting the keep-ratio to be 0.5 achieves a reasonable trade off while having a keep-ratio of 0.25 is too aggressive, which results in a dramatic accuracy drop (3.04%) of the model. We also explore different keep-ratio methods by linearly increasing or decreasing the keep-ratios from early to late layers while keeping their average keep-ratio to be 0.5. Specifically, we use keep-ratios of $[0.25, 0.32, 0.39, 0.46, 0.53, 0.60, 0.68, 0.75]$ (increasing) or its inverse (decreasing) on 8 transformer blocks. As shown in Fig. 1 Right, both models perform significantly worse.

We simulate the gradient calculation on ViT-Tiny with 2000 data batches randomly sampled from the ImageNet training dataset. We apply different keep-ratio methods on the same model with the same fixed weights, and calculate the correlation (measured by cosine similarity) between the weight gradients of applying SBP and the original exact weights gradients, *i.e.*, $Cosine(dW_{SBP}, dW_{no-SBP})$. From Fig. 2, we observe that the weight gradients have a stronger correlation when using uniform keep-ratios compared to the case of increasing or decreasing keep-ratios. We believe that the fact that uniform keep-ratios keep the location of the dropped gradients consistency between different layers helps preserve gradient information at these spatial locations. However, non-uniform keep-ratios will drop at different locations between different layers, which adds noise to produce a less accurate estimation of the original gradients, and hence lowers accuracy.

**How to sample the gradient keep mask?** SBP calculates the gradients by only using a subset of feature maps. A key question is how to sample the subset of feature maps to mask the kept gradients. If we fix the spatial locations of the gradient kept indices for every training step, the backward only propagates the valid gradients (non-zeros) on these fixed locations, which forces the model to only learn from fixed partial spatial information and therefore severely hurts the model performance. A better strategy is to randomly sample the gradient kept indices in each training step, so that it can statistically visit every spatial location with equal importance.

We examine two mask sampling strategies: grid-wise mask sampling and random mask sampling. See Fig. 3 left as a particular example of keep-ratio 0.5. We report the accuracy in Fig. 4 left. Overall, grid-wise mask sampling achieves $0.5\% \sim 1.7\%$ higher accuracy over random mask sampling. In Fig. 3, a stronger correlation is also observed by using the grid-wise sampling. Grid-wise sampling more structured sampling approach seems to provides a more accurate estimation of gradients. Unless stated otherwise, we use grid-wise sampling in this paper.

**How to apply SBP on MHSA?** In general, the transformer block consists of two sub-blocks: Multi-Head Self-Attention (MHSA) and Multi-Layer Perception (MLP). Theoretically, the memory usage of applying SBP on MLP (two linear layers) is proportional to the keep-ratio $r$. However, the memory saving of applying SBP on MHSA depends on the sampling method on the query, key, value (QKV), and the ratio of head dimension $d$ over the number of tokens $n$. From [3], activation maps needed

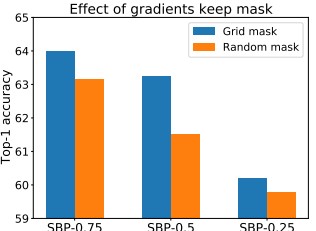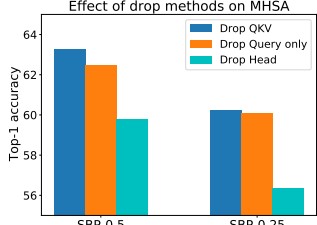

Figure 4: Left: effect of different sampling method for gradient keep mask. Right: effect of different SBP dropping methods on Multi-Head Self-Attention (MHSA).

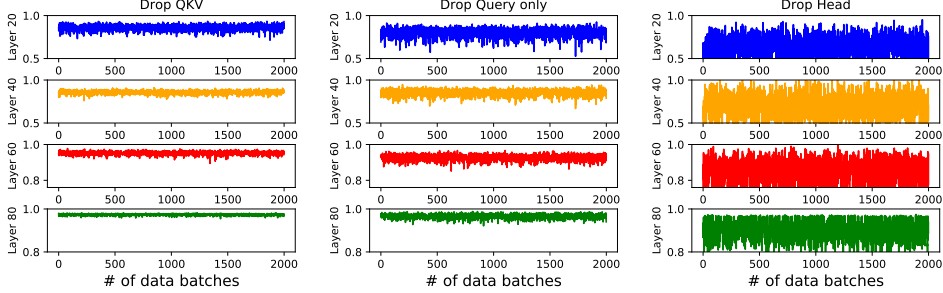

Figure 5: Cosine similarity of weights gradient for different method on applying SBP on ViT-Tiny MHSA layers with a keep-ratio of 0.5. Left: Drop QKV, middle: Drop Query only, right: Drop Head.

for gradients calculation in MHSA include the inputs and outputs ($2hdn$), QKV vectors (each with $hdn$), the attention weight maps ($2hnn$ before and after the softmax), thus requiring the memory of $3hdn + 2hnn$ in total, where $h$ is the number of heads. Dropping gradients on query only (Drop Query only) or on all QKV (Drop QKV) enjoys the following memory usage ratio:

$$\text{Drop Query only:} \quad \frac{2hdn+hdnr+2hnnr}{3hdn+2hnn} = \frac{r(\frac{d}{n}+2)+2\frac{d}{n}}{(\frac{d}{n}+2)+2\frac{d}{n}}$$
$$\text{Drop QKV:} \quad \frac{3hdnr+2hnnr^2}{3hdn+2hnn} = r\frac{3+2r\frac{d}{n}}{3+2\frac{d}{n}} \tag{13}$$

In video transformers, the number of tokens $n$ is much larger than the head dimension $d$ (i.e., in video Swin Transformer [13] , $n = 392$ and $d = 32$, $\frac{d}{n} = 0.082$) and thus the attention weight maps $2hnn$ comprise most of the memory. Therefore, dropping gradients on query only in the video model is good enough to save memory so that its memory is linearly proportional to the keep ratio $r$ [3].

However, in image task models such as ViT with a typical input size 224 and patch size 16, $n = 14 \times 14 = 196$ and $d = 64$, $\frac{d}{n} = 0.327$. With keep-ratio $r = 0.5$ (or $r = 0.25$), dropping gradients on query alone uses $0.61\times$ (or $0.42\times$) memory while dropping all QKV reduces its memory usage ratio to $0.46\times$ (or $0.22\times$). From Fig. 4, dropping gradients on all QKV also gains better accuracy. We also report the performance of dropping gradients along the attention heads, but we did not observe accuracy improvement. Hence, we apply SBP on all QKV for image-based transformers.

We plot the cosine similarity of the weights gradients of applying SBP and the original exact gradients in Fig. 5. One interesting observation is that Drop QKV has a slightly better correlation to Drop Query only and a much higher correlation to Drop Head. We believe an accurate estimation of the gradients might be an important factor to the model performance trained with SBP, which provides insight for further investigation.

## 5 Generalizability of SBP

We evaluate the generalizability of our proposed SBP on two computer vision benchmarks: image classification on ImageNet [18] and object detection on COCO [10]. In order to do a fair comparison of the same network, we keep the training hyper-parameters of optimizer, augmentation, regularization, batch size and learning rate the same for a given model, and only adopt different stochastic depth

Table 1: Accuracy and memory results of applying SBP for ViT and ConvNeXt on ImageNet.

| Network | Keep-ratio | Batch size | Memory (MB / GPU) | Top-1 accuracy (%) |
|---------|-----------|-----------|-------------------|--------------------|
| ViT-Tiny | no SBP | 256 | 8248 | 73.68 |
| ViT-Tiny | 0.5 | 256 | 5587 (0.68×) | 73.09 (-0.59) |
| ViT-Base | no SBP | 64 | 10083 | 81.22 |
| ViT-Base | 0.5 | 64 | 7436 (0.74×) | 80.62 (-0.60) |
| ConvNeXt-Tiny | no SBP | 128 | 12134 | 82.1 |
| ConvNeXt-Tiny | 0.5 | 128 | 7059 (0.58×) | 81.61 (-0.49) |
| ConvNeXt-Base | no SBP | 64 | 14130 | 83.8 |
| ConvNeXt-Base | 0.5 | 64 | 8758 (0.62×) | 83.27 (-0.53) |

augmentation [9] for different model sizes. The Mixed Precision Training [17] method is enabled for faster training. All experiments are conducted on machines with $8\times$ Tesla 16GB V100. Each accuracy is reported with an average result of three runs with different random seeds. For more details of experimental settings, please refer to the supplementary material.

## 5.1 Classification on ImageNet

We report the memory and accuracy trade-off in Tab. 1 for two state-of-the-art networks: transformer-based ViT [6] and convolutional-based ConvNeXt [14]. We train 300 epochs for both these two networks by following the training recipe of ConvNeXt [14].

For ViT models, we apply SBP on the last 8 transformer blocks including MHSA and MLP layers with the keep-ratio of 0.5, reducing GPU memory usage by approximately 30%. Note that the memory saving ratio of the network is not proportional the the drop-ratio ($1-$ keep-ratio) because we only apply SBP on $\frac{2}{3}$ of all the layers. With keep ratio of 0.5, the SBP technique can still effectively learn with an accuracy drop of only 0.59% and 0.60% for ViT-Tiny and ViT-Base, respectively.

For ConvNeXt models, it builds the basic block with a depth-wise convolutional (DW-Conv) layer with kernel size $7\times 7$ followed by two point-wise convolutional (PW-Conv) layers[3] and the DW-Conv layer only consumes a small portion of GPU memory (less than 10%). It has multi-scale stages where early stages have higher redundancy and occupy most of the GPU memory. Therefore, we apply SBP on the PW-Conv and down-sampling layers in the first three stages, and do not apply it on the last stage. With a keep ratio of 0.5, ConvNeXt-Tiny and ConvNeXt-Base can efficiently learn the model with a GPU memory saving of above 40% and an accuracy drop within 0.53%.

## 5.2 Object Detection and Segmentation on COCO

We fine-tune the vanilla Mask R-CNN [8] and Cascade Mask R-CNN [1] on the COCO dataset [10] with ConvNeXt-Tiny and ConvNeXt-Base backbones pretrained on ImageNet-1K, respectively. Following the same hyper-parameter settings of [14], we apply SBP on the ConvNeXt backbones to save training memory. Results in Tab. 2 show that SBP with keep-ratio of 0.5 can still learn the detection task reasonably well with 0.2% and 0.7% loss on the box and mask average precision (AP) for ConvNeXt-Base and ConvNeXt-Tiny backbones, respectively. However, it only consumes about $0.7\times$ of the GPU memory.

Note that training detection models on high resolution images (up to $800\times 1333$) is memory intensive even for a very small batch size. For example, with backbone of ConvNeXt-Base and batch size of 2, the model training requires 17.4GB of GPU memory which may lead to out of memory error for common devices such as Tesla 16GB V100. While with the help of SBP, the model can still be effectively trained on GPUs with 12.5GB memory, which is much more feasible for many research groups. Here we apply SBP only in the backbone, we believe it has a potential to apply SBP on more other layers in the detection model to achieve more promising results in terms of memory efficiency.

---

[3]The two PW-Conv layers are equivalent to an MLP block in transformers.

Table 2: COCO object detection and segmentation results using Mask-RCNN with backbone ConvNeXt-T and Cascade Mask-RCNN with backbone ConvNeXt-B.

| Backbone | Keep-ratio | Batch size | Memory (GB / GPU) | $AP^{box}$ | $AP^{box}_{50}$ | $AP^{box}_{75}$ | $AP^{mask}$ | $AP^{mask}_{50}$ | $AP^{mask}_{75}$ |
|---|---|---|---|---|---|---|---|---|---|
| ConvNeXt-T | no SBP | 2 | 8.6 | 46.2 | 67.9 | 50.8 | 41.7 | 65.0 | 44.9 |
| ConvNeXt-T | 0.5 | 2 | 5.9 (0.69×) | 45.5 | 67.4 | 50.1 | 41.1 | 64.4 | 44.1 |
| ConvNeXt-B | no SBP | 2 | 17.4 | 52.7 | 71.3 | 57.2 | 45.6 | 68.9 | 49.5 |
| ConvNeXt-B | 0.5 | 2 | 12.5 (0.72×) | 52.5 | 71.3 | 57.2 | 45.4 | 68.7 | 49.2 |

## 6   Discussion

In this work, we present a comprehensive study of the Stochastic Backpropagation (SBP) mechanism with a generalized implementation. We analyze the effect of different design strategies to optimize the trade-off between accuracy and memory. We show that our approach can reduce up to 40% of the GPU memory when training image recognition models under various deep learning backbones.

**Limitations:** In general, SBP is a memory efficient training method, but it still causes slight loss of accuracy. In our current results, SBP produces only a small amount of training speedup($\sim 1.1\times$). How to further speed up the training process is part of our future work.

**Societal impact:** Our work studies the stochastic backpropagation mechanism, which can be used for training general deep learning models. Misuse can potentially cause societal harm. However, we believe our work is a general approach and is consist with the common practice in machine learning.

## 7   Acknowledgments and Disclosure of Funding

We thank the anonymous reviewers for their helpful suggestions. This work was funded by Amazon.

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
