# An In-depth Study of Stochastic Backpropagation

**Jun Fang**  **Mingze Xu**  **Hao Chen**  **Bing Shuai**  **Zhuowen Tu**  **Joseph Tighe**
AWS AI Labs
{junfa, xumingze, hxen, bshuai, ztu, tighej}@amazon.com

## 8  Supplementary material

In this supplementary material, we expand our discussion on Stochastic Backpropagation (SBP) with additional analysis and experiments. In particular, we discuss the following:

- Section 8.1 derives the gradient calculation for attention layers.
- Section 8.2 discusses the chain rule effect on the transformer blocks.
- Section 8.3 reports the details of the experiment settings.
- Section 8.4 investigates the insights on the gradient keep-ratios and gradient keep masks on a very deep network ConvNeXt-Base.
- Section 8.5 discusses the vanishing gradient problem.
- Section 8.6 compares the model similarity between with and without applying SBP.

### 8.1  Gradient Calculation of Attention Layers

In section 3.2, we provide the gradient calculation of linear layers (or PW-Conv) and general convolutional layers for the backward phase of SBP. Here we derive the equations to the attention layers, as they are the important layers in the transformer-based architectures [6, 11].

In the original multi-head self-attention (MHSA) module of vision transformers [6], given a layer $i$, it first linearly transforms the input tensor $X_i \in \mathbb{R}^{B \times N_i \times C_i}$ (flatten from feature map $X_i \in \mathbb{R}^{B \times T_i \times H_i \times W_i \times C_i}$) to query tensor $Q_i \in \mathbb{R}^{(B \times N_i) \times (h \times d_{Q_i})}$, key tensor $K_i \in \mathbb{R}^{(B \times N_i) \times (h \times d_{K_i})}$, value tensor $V_i \in \mathbb{R}^{(B \times N_i) \times (h \times d_{V_i})}$ by linear layers with learnable weights $W_{Q_i} \in \mathbb{R}^{C_i \times (h \times d_{Q_i})}$, $W_{K_i} \in \mathbb{R}^{C_i \times (h \times d_{K_i})}$, $W_{V_i} \in \mathbb{R}^{C_i \times (h \times d_{V_i})}$. The query $Q_i$, key $K_i$, value $V_i$ and their corresponding weights $W_{Q_i}, W_{K_i}, W_{V_i}$ are reshaped as a matrix format. The notation $N_i = T_i \times H_i \times W_i$ is the number of tokens, $h$ is the number of heads, and $d_{Q_i}, d_{K_i}$ and $d_{V_i}$ are the dimensions of query, key and value, respectively. The forward pass of these three linear mapping is

$$Q_i = X_i W_{Q_i}, \quad K_i = X_i W_{K_i}, \quad V_i = X_i W_{V_i}. \tag{14}$$

It then calculates a scaled dot-product attention,

$$M_i = \frac{Q_i K_i^T}{\sqrt{d_{K_i}}}, \quad S_i = \text{softmax}(M_i),$$

$$X_{i+1} = A_i = \text{Attention}(Q_i, K_i, V_i) = \text{softmax}\left(\frac{Q_i K_i^T}{\sqrt{d_{K_i}}}\right) V_i = S_i V_i \tag{15}$$

During the traditional backward pass with full backpropagation, the gradients of query, key, value are calculated by

$$dQ_i = dM_i \frac{K_i}{\sqrt{d_{K_i}}}, \quad dK_i = \frac{Q_i^T}{\sqrt{d_{K_i}}} dM_i, \quad dV_i = S_i^T dA_i. \tag{16}$$

The gradients of weights $W_{Q_i}, W_{K_i}, W_{V_i}$ are as follows

$$dW_{Q_i} = X_i^T dQ_i, \quad dW_{K_i} = X_i^T dK_i, \quad dW_{V_i} = X_i^T dV_i, \tag{17}$$

and the gradient of input $X_i$ is

$$dX_i = dQ_i W_{Q_i} + dK_i W_{K_i} + dV_i W_{V_i}. \tag{18}$$

To apply SBP on the MHSA layers, we apply gradients dropout on the attention map $M_i$ (has dimension $hN_i^2$) instead of the feature map $X_{i+1}$ (has dimension $hd_{V_i}N_i$) because the attention map $M_i$ dominants the memory usage as $N_i > d_{V_i}$. We refer to section 4 and Eq. (13) for more details of the memory usage.

We also have different choices of dropping gradients and here we give an example of dropping on query only, other methods such as dropping on heads or dropping on all QKV can be easily derived in a similar way. Similar to the analysis of linear layer case in section 3.2.1, we split the query tensor into two subsets $Q_i^{keep}$ and $Q_i^{drop}$ along the spatial dimension, and the forward spatial nodes in query tensor $Q_i$ and the attention weights maps $M_i$ can be calculated independently as:

$$\begin{aligned}
Q_i^{keep} &= X_i^{keep} W_{Q_i}, \quad Q_i^{drop} = X_i^{drop} W_{Q_i}, \\
M_i^{keep} &= \frac{Q_i^{keep} K_i^T}{\sqrt{d_{K_i}}}, \quad M_i^{drop} = \frac{Q_i^{drop} K_i^T}{\sqrt{d_{K_i}}}.
\end{aligned} \tag{19}$$

Regarding to the backward, SBP drops gradients with superscripts $^{drop}$ on $Q_i$ and $M_i$. It sets $dM_i^{drop} = \mathbf{0}$ and has

$$\begin{aligned}
dQ_i &= [dQ_i^{keep}, dQ_i^{drop}], \\
dQ_i^{keep} &= \frac{1}{\sqrt{d_{K_i}}} dM_i^{keep} K_i^T, \\
dQ_i^{drop} &= \frac{1}{\sqrt{d_{K_i}}} dM_i^{drop} K_i^T = \mathbf{0},
\end{aligned} \tag{20}$$

and

$$\begin{aligned}
dK_i &= \frac{1}{\sqrt{d_{K_i}}} Q_i^T dM_i = \frac{1}{\sqrt{d_{K_i}}} [Q_i^{keep}, Q_i^{drop}]^T [dM_i^{keep}, dM_i^{drop}], \\
&= \frac{1}{\sqrt{d_{K_i}}} (Q_i^{keep^T} dM_i^{keep} + Q_i^{drop^T} dM_i^{drop}) = \frac{1}{\sqrt{d_{K_i}}} Q_i^{keep^T} dM_i^{keep}.
\end{aligned} \tag{21}$$

That is, the query $Q_i$ has exact gradients on the *kept* locations and zero gradients on the *dropped* locations, and the gradient of key $dK_i$ has neither exact nor zero gradients but estimated by $\frac{1}{\sqrt{d_{K_i}}} Q_i^{keep^T} dM_i^{keep}$.

Since there is no gradient dropout on the value tensor $V_i$, from Eq. (17), it's gradient $dV_i$ will not be affected by SBP and will be the same to the gradient (we call it the exact gradient) of the model without applying SBP.

However, the gradients of query and key weights will be updated and not be the same to the original case without applying SBP. More specifically, we have the gradients of query weights

$$\begin{aligned}
dW_{Q_i} &= X_i^T dQ_i = [X_i^{keep}, X_i^{drop}]^T [dQ_i^{keep}, dQ_i^{drop}] \\
&= X_i^{keep^T} dQ_i^{keep} + X_i^{drop^T} dQ_i^{drop} = X_i^{keep^T} dQ_i^{keep}
\end{aligned} \tag{22}$$

and the gradients of key weights

$$dW_{K_i} = X_i^T dK_i = \frac{1}{\sqrt{d_{K_i}}} X_i^T Q_i^{keep^T} dM_i^{keep}. \tag{23}$$

They are all calculated as an approximated version of their original gradients by only using the feature maps at the *kept* indices. From Eq. (20) and (21), we can also update the gradients of input $X_i$ in Eq. (18) and it will be an approximated version of its original case as well.

## 8.2 Chain Rule Effect of Transformer Blocks

In general, transformer block consists of two sub-blocks: Multi-Head Self-Attention (MHSA) and Multi-layer perception (MLP). The MLP sub-block is equivalent to two PW-Conv or linear layers. In section 3.3.1, we discuss the chain rule effect on two consecutive PW-Conv layers $f_i$ and $f_{i-1}$,

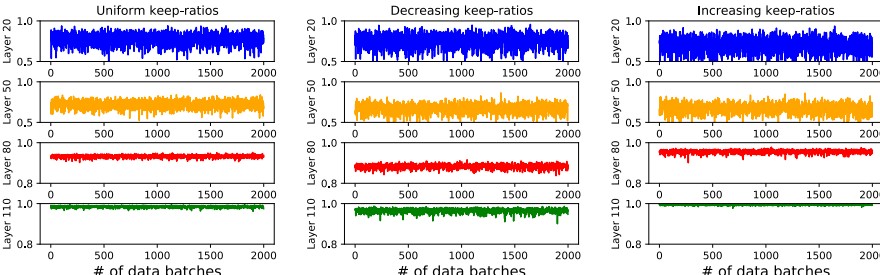

Figure 6: Cosine similarity of weights gradient between with and without applying SBP on ConvNeXt-Base with uniform, decreasing, and increasing keep-ratios.

here we extend the chain rule effect to the previous MHSA layers $f_{i-2}$. From section 3.3.1 and 4 (Fig. 1 right), using uniform keep-ratios and having the same gradients dropping indices set on consecutive layers can preserve more gradient information and gain higher accuracy, thus in this section we only consider the same keeping and dropping indices on the entire transformer block. That is, $\mathbb{Z}_{i+1}^{keep} = \mathbb{Z}_i^{keep} = \mathbb{Z}_{i-1}^{keep}$ and $\mathbb{Z}_{i+1}^{drop} = \mathbb{Z}_i^{drop} = \mathbb{Z}_{i-1}^{drop}$.

From 3.3.1, $dX_{i-1}$ has non-zero gradients (which are exact to the original case without SBP) at *kept* indices and zero gradients at *dropped* indices. By chain rule, the backward gradients pass to the MHSA layer output $A_{i-2}$ and we have $A_{i-2}^{keep}$ to be exact gradients and $A_{i-2}^{drop} = \mathbf{0}$. From the Eq. (15), (16), and (17), the gradients of value tensor $dV_{i-2}$ and value weights $dW_{V_{i-2}}$ will be all affected and no longer be the exact. Therefore, the gradients of all QKV weights and activations will be updated and calculated as an estimation to the original gradients of mini-batch SGD without applying SBP.

## 8.3  Experimental Settings

In this section, we report the experimental details of training settings. For ImageNet training of both ViT and ConvNeXt, we follow the same hyper-parameter settings of Table. 5 in [14] except that we use different stochastic depth rates for different models. Specifically, we set stochastic depth rates 0.0, 0.5, 0.1, 0.5 (and 0.0, 0.3, 0.1, 0.3) for ViT-Tiny, ViT-Base, ConvNeXt-Tiny, ConvNeXt-Base without applying SBP (and with applying SBP with a keep-ratio of 0.5), respectively. For ViT-Tiny and ViT-Base, we apply SBP on the last 8 transformer blocks. For ConvNeXt-Tiny and ConvNeXt-Base, we apply SBP on all blocks of the first two stages. On the third stages, we apply SBP on the first 6, 21 blocks for ConvNeXt-Tiny, ConvNeXt-Base, respectively. We use one machine (each machine has $8\times$ Tesla 16GB V100) to train ViT-Tiny and ConvNeXt-Tiny and 4 machines to train ViT-Base and ConvNeXt-Base.

For COCO experiments, we follow the same training settings used in Section A.3. of [14]. We only apply SBP on the ConvNeXt backbones. We use the backbone weights pre-trained from ImageNet as network initializations. We use one machine to train the detection task.

## 8.4  Gradient Keep-ratios and Keep Masks on ConvNeXt-Base

We further investigate the insights on the gradient keep-ratios and gradient keep masks on a very deep network, ConvNeXt-Base, which has more than 100 layers. We plot the cosine similarity of weights gradient between with and without applying SBP on layer 20, 50, 80, 110 of ConvNeXt-Base, which are PW-Conv1, DW-Conv, PW-Conv2, DW-Conv layers, respectively.

From Fig. 6, we also observe that the uniform keep-ratios method has an overall stronger correlation compared to non-uniform keep-ratios methods. More specifically, compared to uniform keep-ratios method, decreasing keep-ratios method has a weaker correlation on deeper layers as the keep-ratios are smaller in the deeper layers. Although the increasing keep-ratios method enjoys a stronger correlation on deeper layers, it has a smaller gradient keep-ratio as well as a weaker correlation in early layers. This observation is consistent between the ViT and ConvNeXt networks.

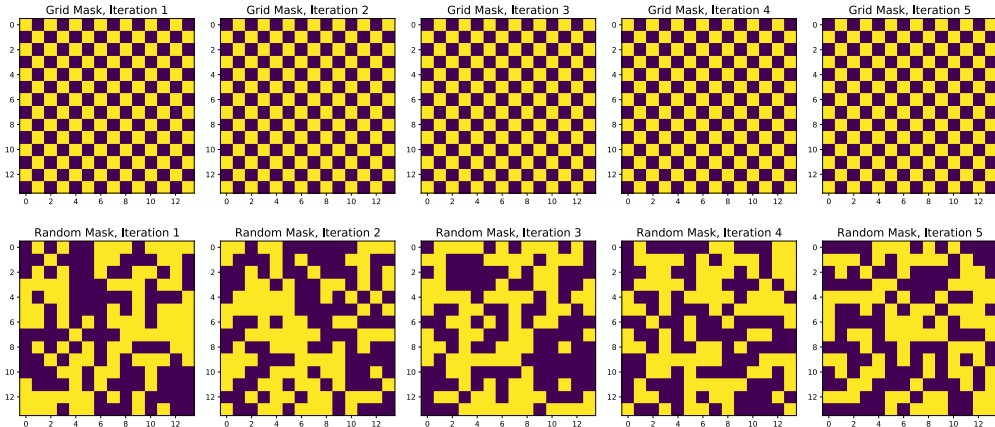

Figure 7: A particular example of sampled gradient keep masks for 5 iterations. Keep-ratio is 0.5. Upper: grid-wise sampling method. Lower: random sampling method.

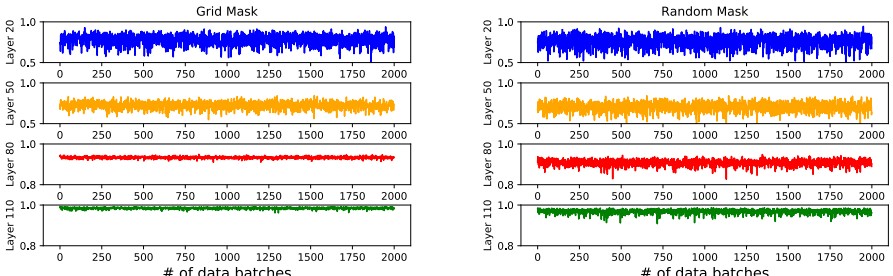

Figure 8: Cosine similarity of weights gradient between with and without applying SBP-0.5 on ConvNeXt-Base with grid-wise sampling and random sampling method.

Next, we compare the grid-wise sampling mask and random sampling mask. In the training process, we randomly sample the gradient keep mask in every iteration so that every spatial location will be statistically visited with equal importance. Once the mask is sampled in each iteration, we apply uniform keep-ratios for every SBP layer with the same gradient keep mask. Fig. 7 gives a particular example of the sampled masks for 5 iterations with a keep-ratio of 0.5. In the first two iterations (upper left of Fig. 7), the grid-wise sampling can visit all spatial locations, which is not observed in the random sampling. In Fig. 8, it is also observed that the grid-wise sampling achieves a stronger correlation compared to the random sampling. In general, the correlation behaviors of gradient keep-ratios methods and gradient sampling methods are consistent to both ViT and ConvNeXt networks.

## 8.5 Vanishing Gradients?

We plot the L2 norm of weight gradients of some or all layers of the model during training process for ViT-Tiny (Fig. 9) and ConvNeXt-Base (Fig. 10). Although SBP discards some parts of gradient information, especially on activations, we did not observe the vanishing gradient problem on weights. However, from (10) and (11), in an extreme case that two consecutive SBP layers have non-overlapped *keep* indices, i.e., $\mathbb{Z}_{i+1}^{keep} \cap \mathbb{Z}_{i}^{keep} = \emptyset$, the gradient information on all indices will be dropped, and therefore, the vanishing gradient occurs. We point out that this is never the case in practice and it can be simply avoided by using the same gradient keep mask across all SBP layers.

## 8.6 Model Similarity

We plot the cosine similarity of model weights in Fig. 11 to show the change curve during the entire training process. The training hyper-parameters and model initialization weights are the same for model trained with and without applying SBP. We observe that the cosine similarity of model weights

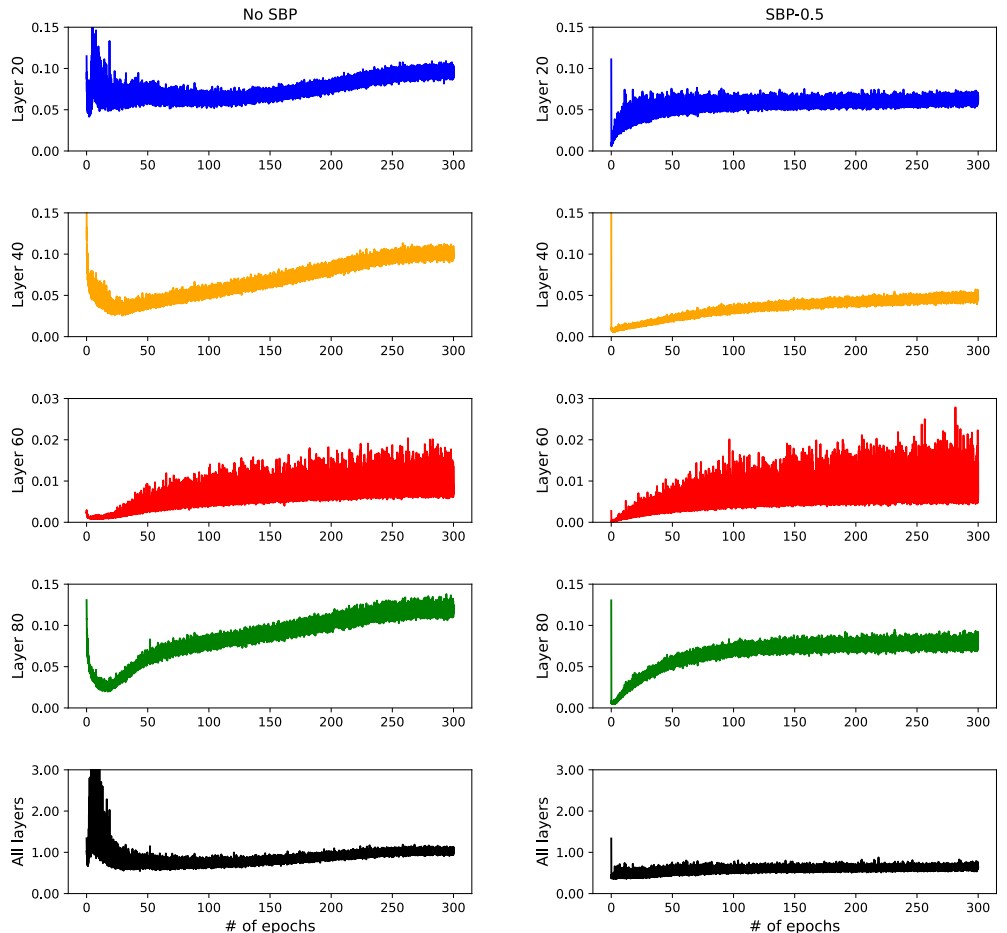

Figure 9: L2 norms of weights gradient on some or all layers of ViT-Tiny. Left: baseline without applying SBP. Right: with applying SBP-0.5.

gradually decay from 1.00 to 0.14 for ViT-Tiny and 0.06 for ConvNeXt-Base, which indicates that the final model weights may not be similar after the dynamic training process. One explanation is that at the early stage of training, model weights with SBP only show a small difference compared to those without SBP. However, as the the training goes further and further, this small difference will be propagated and become larger and larger.

We also evaluate the sample-level top-1 prediction consistency rate between models trained with and without applying SBP. For ViT-Tiny and ConvNeXt-Base cases, the consistency rates on ImageNet validation dataset are 80.73% and 91.12%, respectively. The consistency rates are much higher than their corresponding top-1 accuracies, which demonstrates that models trained with or without applying SBP can can take different learning paths and end up with different local minima, but with similar prediction performance.

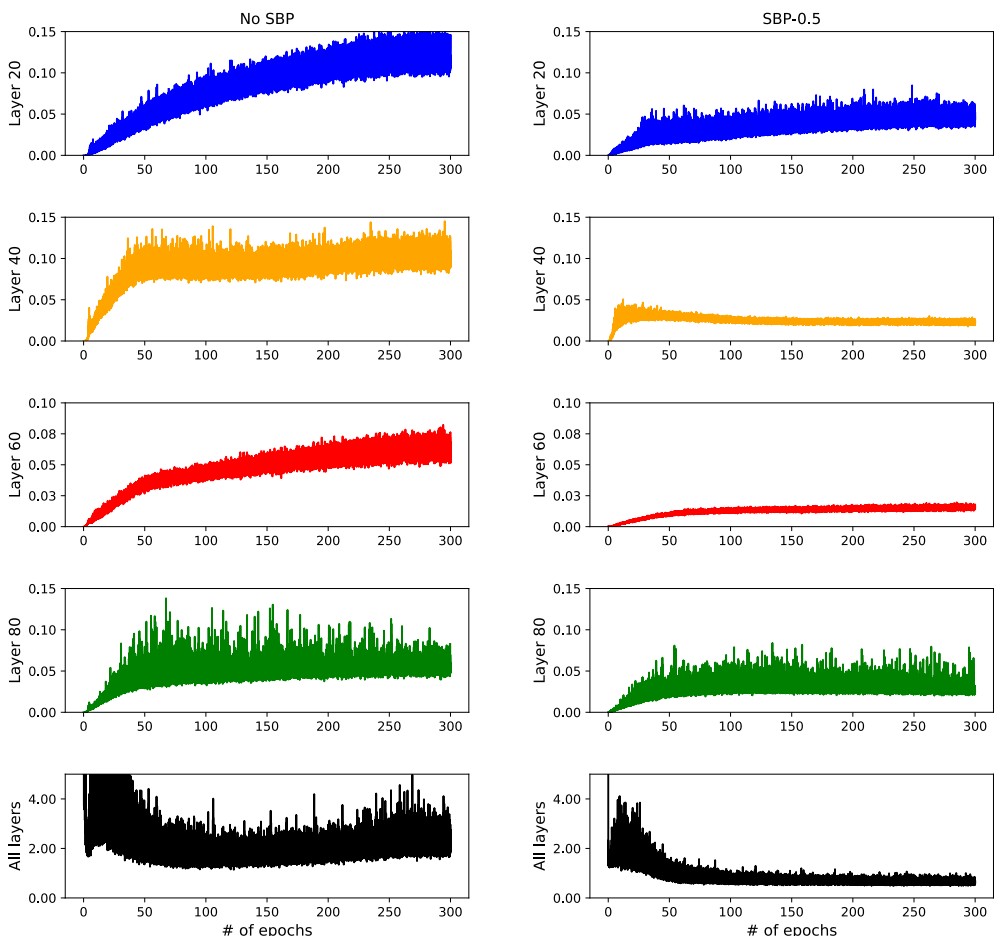

Figure 10: L2 norms of weights gradient on some or all layers of ConvNeXt-Base. Left: baseline without applying SBP. Right: with applying SBP-0.5.

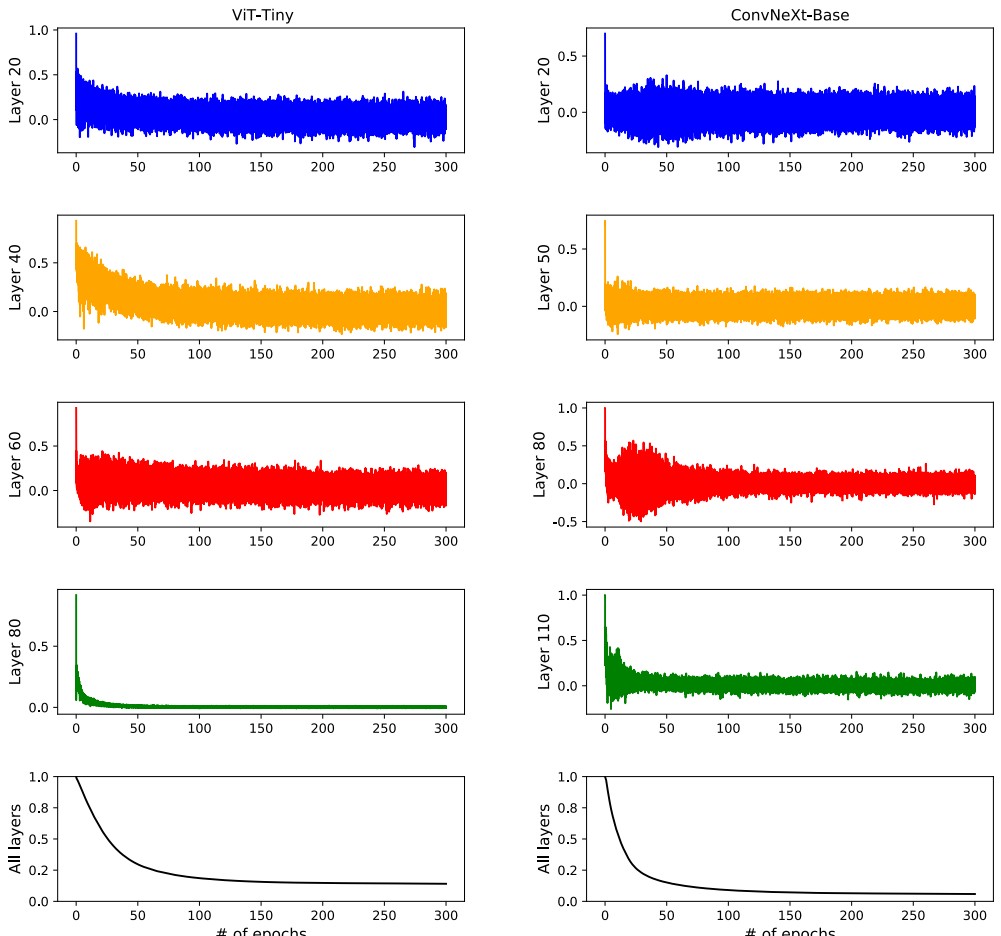

Figure 11: Cosine similarity of model weights on some or all layers between models with and without applying SBP-0.5.