# OpenReview forum: "An In-depth Study of Stochastic Backpropagation"
_NeurIPS.cc/2022/Conference — NeurIPS 2022 Accept_

### Official Review · Reviewer_ySTN · 2022-07-13

**Rating:** 5
**Confidence:** 4
**Soundness:** 3 good
**Presentation:** 2 fair
**Contribution:** 3 good

**Summary:**

A study of SBP, applied to image classification and object detection, is provided. A significant amount of memory is saved during training while the degradation of accuracy is relatively small. Derivations and experimental results are comprehensive and they validate the author's main points.

**Questions:**

As I pointed out in the weaknesses part, I want to know more about novelty (compared to CVPR `22 paper) and technical contribution (what was the most difficult technical problem?).

**Limitations:**

Limitations are adequately addressed.

**Strengths And Weaknesses:**

The main strength of the proposed method is its practicality. It saves memory with almost no harm to the final models of interest. There is no reason someone faced with memory limitation not try this simple approach. Derivations related to stochastic backprop. are comprehensive and most of them look sound but I am not fully confident.

I have checked the details of experimental results and ablation studies, and as far as I understand they are convincing. For example, Figure 3 (grid mask vs random mask) experiment matches the theory from the derivation.

Three main weaknesses are novelty, technical contribution, and writing. First, the concept of stochastic backprop. is introduced in the original CVPR paper and the proposed method in the paper is incremental to the original paper. Second, it is not easy to find a non-trivial technical challenge introduced, tackled, and solved by the paper. Third, writing is difficult to follow. I have listed some of the parts that can be improved:
L77 fully connection layer -> fully-connected layer
L84 caching of feature maps dominants the memory usage. -> did not understand
L311 slightly loss -> slight loss

---

> ### Author Response · Authors · 2022-08-02
> **Response to Reviewer ySTN**
>
> Thank you for your valuable feedback. We are encouraged that you find our method simple and practical, and our derivations comprehensive. We address your concerns below:
>
> (1) “novelty and technical contribution”
>
> **R**: Our main contribution is not to introduce the concept of SBP, but to provide theoretical and empirical groundwork for a broader scope of SBP. We also provide insights, guidances, and comprehensive analysis to support the benefits of SBP and argue that more widespread usage can be expected from SBP. We also offer a simple implementation and practical design choices of SBP, which can be a strong starting point for further study.
>
> More specifically, our main contribution is providing a comprehensive study of SBP in the following aspects:
>
> * Formulate SBP in a boarder scope, which not just focuses on video frames [1] but also extends to feature maps, and provide theoretical analysis to understand how SBP affects the gradient calculation. These are missed in [1].
> * Generalize the SBP technique to more tasks including image classification and object detection, more different networks including ViT and ConvNeXt, and more operators including CNN, etc. In comparison, [1] only focused on video models.
> * Provide insights and guidance on a good design strategy for SBP, while the design strategy in [1] is adhoc without guidances. A naive extension of [1] to image tasks will have severe accuracy degradation (more than 10% of top-1 accuracy drop on ImageNet).
> * Present a very simple and efficient implementation of the method in Algorithm 1, which does not require any re-computation of nodes during backward. However, one extra pass of re-forward computation is required in [1].
>
> [1] Feng Cheng, Mingze Xu, Yuanjun Xiong, Hao Chen, Xinyu Li, Wei Li, and Wei Xia. Stochastic backpropagation: A memory efficient strategy for training video models. CVPR, 2022.
>
>
> (2) “writing is difficult to follow”
>
> **R**: We apologize that there are typos and grammatical mistakes in the original version. We had multiple rounds of proofreading to correct inconsistencies, typos and errors in the revision. Please refer to our response to the *Reviewer WMoQ* for a detailed list of corrections we have made.

---

> > ### Comment · Reviewer_ySTN · 2022-08-08
> > **Post-Rebuttal Score Change**
> >
> > Dear authors, other reviewers and chairs,
> >
> > Initially, I thought the proposed method was a too simple and limited technical contribution.
> >
> > After reading the other reviewer's comments and the answers from the authors, I have changed my mind and the final rating to borderline accept from borderline reject.
> >
> > Although this might look like a small change, I am at least inclined towards accepting now.
> >
> >
> > Best regards,
> >
> > Reviewer ySTN.

---

> > > ### Author Response · Authors · 2022-08-08
> > > **Re: Post-Rebuttal Score Change**
> > >
> > > Thank you for your reconsideration and the kind notice about the score change!

---

### Official Review · Reviewer_yhnq · 2022-07-15

**Rating:** 6
**Confidence:** 3
**Soundness:** 3 good
**Presentation:** 4 excellent
**Contribution:** 3 good

**Summary:**

Stochastic Backpropagation (SBP) is a memory efficient training method for training video models. In this work authors presented the in-depth study of stochastic back propagation in generalized way and use this strategy for training deep neural network models for image classification and object detection tasks.

Unlike earlier work of SBP on video models, where frames were during backpropagation however keeping all the frames for forward propagation. The authors in this paper, drop feature maps in earlier layers during backpropagation, keeping all the feature maps in forward propagation phase. This is somewhat conceptually like drop out, however, limited to the backpropagation phase.

The SBP is studied for point-wise convolution layers, convolution layers and transformer layers. SBP approximation behaves differently for different operators. For point-wise convolution SBP approximation of gradients result in 0 gradients for dropped indices for kept indices the gradients are exact in comparison to SGD. This contrasts with convolution layers with kernel size greater than 1, SBP approximation to gradient is inexact and not zero irrespective of the indices dropped or kept except some special scenarios.

In addition to the study of SBP, authors experimented with various design choices example, number of layers on which SBP needs to be applied, kept ratio, dropping strategy.  Based on the experiments, was concluded to apply SBP to 2/3 layers and have 0.5 keep ratio using grid wise sampling. This design strategy gave sufficient GPU memory saving without large drop in accuracy, in addition, the gradients gave higher correlation to SGD gradients without SBP in terms of cosine similarity.

The generalizability of SBP is evaluated on two computer vision benchmarks: ImageNet and object detection COCO. Applying SBP reduced 40% GPU memory with less than 1% drop in accuracy


**Questions:**

Q1) Even though the accuracy drop is less than 1% there are few aspects of SBP that raise question if SBP can be thought as efficient training methodology or altogether different training strategy.
1) Fixed grid sampling for backpropagation is clearly making few features more important than the others.
2) Cosine similarity though gives a promise, i.e., e SBP approximated gradients are correlated with SGD gradients and eventually the trained model might end up in similar final weights. However, how well it extends to the benchmarking tasks of ImageNet and classification is not clear.
May be one way to show SBP as an approximation could be to show if with or without SBP the correct examples are similar.
or if the weights with or without SBP keeping all training parameters constant are close.
Can you please throw light on this ?


**Ethics Review Area:**

["I don’t know"]

**Limitations:**

SBP is memory efficient training method. However, SBP is not exact and result in loss of training accuracy.
SBP specifically when applied to convolution layers  with kernel size greater than 1 and also in query of transformers are neither zero nor exact. The approximations even though result in very minimal loss in accuracy, however, the effect of this approximation on the model in the perspective of of explainibility and reliability requires separate study.

**Strengths And Weaknesses:**

SBP is memory efficient training method. It seems  to generalize well across tasks like image classification and object detection and is very simple and efficient to implement.
SBP provides 40% GPU saving with  ~1.1 x training speed up with less than 1% accuracy drop.
Unlike other training methods like gradient accumulation, SBP is not exact and result in loss of training accuracy.
The paper is very promising , since it gives a training approach which is memory efficient and comparatively faster with equivalent performance.

---

> ### Author Response · Authors · 2022-08-02
> **Response to Reviewer yhnq**
>
>
> Thank you for your insightful comments. We are encouraged that you find our paper very promising. We have the following responses for your questions.
>
> (1) “Fixed grid sampling for backpropagation is clearly making few features more important than the others.”
>
> **R**: Thank you for raising this important question. As we explained in L231 - L232, this is not the case in our design. In each training step, we randomly sample the gradient kept indices in a grid-wise manner so that it can statistically visit every spatial location with equal importance. We show a particular example of how to sample the gradient keep mask in Figure 7 of supplementary material for a better illustration.  However, if we fix spatial locations of the gradient kept indices during the entire training process, the model will force features at these fixed spatial locations to be more important than the others, which may hurt the model performance.
>
> (2) “Cosine similarity -  extends to the benchmarking tasks of ImageNet and classification is not clear.”
>
> **R**: We extend our cosine similarity analysis to the ViT and ConvNeXt networks on ImageNet classification task.  Similar insights are also observed on these more general cases.  Please refer to our response (1) to *Reviewer Jp2J*, Section 4 and Section 7.4 in the revised paper for more details.
>
> (3) “Cosine similarity - the trained model might end up in similar final weights”
>
> **R**: We additionally plot the cosine similarity of model weights in Figure 11 in the updated supplementary material to show the change curve during the entire training process. The training hyper-parameters and model initialization weights are the same. We observe that the cosine similarity of model weights gradually decays from 1.0 to 0.14 for ViT and from 1.0 to 0.06 for ConvNeXt, which indicates that the final model weights may not be similar after the dynamic training process. One explanation of this observation is that at the early stage of training, model weights with SBP only show a small difference compared to those without SBP. However, as the the training goes further and further, this small difference will be propagated and become larger and larger. Models trained with and without SBP can take different learning paths and end up with different local minima, but with similar prediction performance.
>
> (4) “show if with or without SBP the correct examples are similar”
>
> **R**: We evaluate the sample-level top-1 prediction consistency rate between models trained with and without SBP. Results on ImageNet validation dataset are reported in the following table. For ViT-Tiny and ConvNeXt-Base cases, the consistency rates are more than 80% and are much higher than top-1 accuracies, which demonstrates that the model trained with or without SBP can obtain similar sample-level prediction performance.
>
> | Evaluation of model pair on ImageNet | Top-1 accuracy (%) | Top-1 consistency rate (%) |
> |:------------------------------------:|:------------------:|:--------------------------:|
> |       (ViT-Tiny, ViT-Tiny-SBP)       |    (72.68, 72.64)   |            80.73           |
> |  (ConvNeXt-Base, ConvNeXt-Base-SBP)  |    (83.80, 83.32)   |            91.12           |
> |                                      |                    |                            |
>
> (5) “ConvLayer with kernel size > 1 and query in transformer, gradients are neither zero nor exact. ”
>
> **R**: As derived in Section 3.2.2 and Section 7.1, the gradient calculation when applying SBP on ConvLayer with kernel size > 1 and transformer layer will end up with being neither zero nor exact. However, their gradients still have a strong correlation to the original exact gradients, which are observed from the cosine similarity plots of weights on ViT (contains transformer layer) and ConvNeXt (contains ConvLayer with kernel size 7x7) in Figure 5 and Figure 8, respectively. This observation indicates that approximated gradients by SBP may have a potential to effectively learn the model.
>
> (6) “the effect of this approximation on the model in the perspective of of explainability and reliability requires separate study.”
>
> **R**: Thanks for this interesting idea. The explainability and reliability of this approximation effect definitely deserve further study. For example, can we theoretically prove that this approximation of gradients is always highly correlated to the exact gradients? Under what conditions can this approximation guarantee the convergence of training? Though these questions are beyond the scope of this paper, they will be our future research topics.

---

> > ### Comment · Reviewer_yhnq · 2022-08-10
> > **Great works Thanks**
> >
> > Thank you for detailed response and adding the details in supplementary material

---

> > > ### Author Response · Authors · 2022-08-10
> > > **Re: Great works Thanks**
> > >
> > > We appreciate the reviewer taking time and effort to further review our response!

---

### Official Review · Reviewer_Jp2J · 2022-07-21

**Rating:** 8
**Confidence:** 2
**Soundness:** 4 excellent
**Presentation:** 4 excellent
**Contribution:** 4 excellent

**Summary:**

This paper evaluates the concept of Stochastic Backpropagation (SBP) on object detection and image classification using a traditional convolutional neural network backbone (ConvNeXt) and a transformer backbone (ViT). This expands on the work of [4] that introduced the concept of SBP exclusively for video inputs. The paper provides a formalization of the topic, evaluates efficient implementations, and examines design strategies in determining how to best apply SBP. Quantitative performance is evaluated on the widely used ImageNet and COCO datasets.

**Questions:**

* A major difficulty in analyzing any training algorithm that differs from layer to layer is that there too many different ways to structure the algorithm to exhaustively examine every one. The design experiments in this paper only look at a 4-layer network (L204). How well might these insights generalize to deeper networks, especially those with skip connections?

* When using SBP, are precautions necessary to prevent vanishing gradients? It seems that any time gradient information is discarded, the network runs the risk of catastrophically interrupting the flow of information.

**Limitations:**

The paper addresses a small set of limitations (decrease in accuracy) in Section 6.

**Strengths And Weaknesses:**

__Technical Contribution__

This paper demonstrates the widespread applicability to vision tasks of a reasonably straightforward concept: improved memory efficiency during network training by stochastically choosing some subset of layers to compute gradients for at each backpropagation pass. This work builds on the work of [4], providing a theoretical and empirical groundwork for broader usages than the video prediction use case of [4].  Although this work did not introduce the concept of SBP, it provides a very good argument that SBP should see more prominent usage.

The efficient implementation proposed in Algorithm 1 is clearly demonstrated, and it is explained that it does not require any recomputation of values required for backpropagation.  The design analysis in Section 3 is useful context for understanding the impact that design choices will have (e.g. Figures 1-4). However, the combinatorially explosive number of ways SBP could be applied to a network raises questions about how generalizable these insights are. Nevertheless, specific outcomes like grid sampling outperforming random sampling when choosing gradients to keep, as shown in Figures 3 and 4, are insightful.


__Presentation__

This paper sets up the problem really well in L25-36. It makes very clear how [4] provides a limited scope of evaluation and then indicates exactly how it plans to expand on the scope of [4]. Without even leaving the first page, it is clear to the reader what this paper is about.

One odd convention that stood out was the inclusion of Related Work as Section 5, all the way on the last page of the paper. This paper does an effective job setting the board context for SBP in the introduction (L13-24), but having Related Work as the second section is usually helpful for a reader to compare and contrast the introduced topic to prior work. This contextualizes the information they are about to consume. The authors might consider shifting Section 5 as is up to follow the introduction directly.

There are a couple grammar and spelling errors throughout the work (e.g. L212 "loss" should be "lose"), but nothing too glaring. It would be worth reviewing the paper for such errors before any final submission.

__Summary__

This paper demonstrates impressive results using SBP for vision tasks. The quantitative performance are enormous (Tables 1 and 2, 25-40% memory reduction with less than 1% accuracy loss) on two commonly used backbones for typical vision tasks. The work also demonstrates a simple implementation, which suggests that the concept can become a effective widespread idea. The in depth analysis of design choices also provides future researchers with a strong starting point for further study. While I am admittedly not well read in this line of research, this paper does not appear to have many flaws, generally. It is well-structured and easy to understand, and it provides strong empirical support for the benefits of SBP. Moreover, it asks the right questions about SBP and provides solid initial answers. I think this work is a welcomed contribution to NeurIPS.

---

> ### Author Response · Authors · 2022-08-02
> **Response to Reviewer Jp2J**
>
> Thank you for your constructive feedback. We appreciate that you find our results impressive, our idea effective widespread, our paper well-structured and easy to understand. We address your concerns below.
>
> (1) “How well might these insights generalize to deeper networks, especially those with skip connections?”
>
> **R**: We have verified the generalizability of these insights on two deeper networks, including ViT-Tiny (more than 80 layers) and ConvNeXt-Base (more than 100 layers). We updated the results in Figures 2, 3, 5 of the main paper for Vit-Tiny and Figures 6, 8 of the supplementary material for ConvNeXt-Base. These two networks contain a wide range of layer types including attention layer, CNN, MLP, LayerNorm, and skip connections. Please refer to Section 4 and Section 7.4 for more details. In these more general cases, similar insights are also observed:
>
> * Uniform keep-ratios have a stronger correlation than non-uniform keep-ratios
> * Grid-wise sampling gains a stronger correlation than random sampling for the gradient keep mask
> * Drop QKV enjoys a stronger correlation compared to Drop Query only and Drop Head for the gradient dropping method on the attention layer
>
> |    Network    | # of layers |           $\qquad       \qquad $          Layer types                | Figures in revision | Sections in revision |
> |:-------------:|:-----------:|:------------------------------------------:|:-------------------:|:--------------------:|
> |    ViT-Tiny   |      87     | Attention, MLP,  LayerNorm, SkipConnection |   Figures 2, 3, 5   |       Section 4      |
> | ConvNeXt-Base |     190     |    CNN, MLP,  LayerNorm, SkipConnection    |     Figures 6, 8    |      Section 7.4     |
> |               |             |                                            |                     |                      |
>
>
> (2) “When using SBP, are precautions necessary to prevent vanishing gradients?”
>
> **R**: Thank you for this question and this is a good point. We have additionally plotted the L2 norm of weight gradients of some or all layers of the model during the training process in Figure 9 (ViT-Tiny) and Figure 10 (ConvNeXt-Base) in the revised supplementary material. Although SBP discards some parts of gradient information, especially on the activations, we did not observe the vanishing gradient problem on weights. However, from equation (10) - (11), in an extreme case that two consecutive SBP layers have non-overlapped keep indices, i.e., $Z_{i-1}^{keep} = Z_{i+1}^{keep} \cap Z_{i}^{keep} = \emptyset$, the gradient information on all indices will be dropped, and therefore, the vanishing gradient occurs. We point out that this is not the case in practice and can be simply avoided by using the same gradient keep mask across all SBP layers.
>
> (3) “shifting Section 5 as is up to follow the introduction directly”
>
> **R**: Thank you for the helpful suggestion. We have moved the Related Work Section right after the Introduction Section for a smoother flow of reading.
>
> (4) “a couple grammar and spelling errors”
>
> **R**: We had multiple rounds of proofreading to correct typos, grammatical mistakes and spelling errors in the revision. Please refer to our response to the *Reviewer WMoQ* for more details.

---

> > ### Comment · Reviewer_Jp2J · 2022-08-08
> > **Follow Up**
> >
> > Thank you, authors, for providing answers to the questions I posed in my reviews. I agree that the change of vanishing gradients seems pretty limited in practice. It will be helpful to note this in the paper, specifically the actionable step that can be taken to avoid it altogether.
> >
> > The additional studies are also appreciated, and I agree that they appear to validate the previous experiments done on smaller networks.
> >
> > Looking through the other reviews and reading through the authors' responses to them, I feel confident that my original inclination to recommend acceptance is the right one. This paper will be a nice addition to the conference.

---

> > > ### Author Response · Authors · 2022-08-08
> > > **Re: Follow Up**
> > >
> > > We appreciate your support and the follow-up comment. We will make sure to clarify the vanishing gradient problem in the paper, as well as the practical steps to avoid it.

---

### Official Review · Reviewer_WMoQ · 2022-07-21

**Rating:** 7
**Confidence:** 1
**Soundness:** 3 good
**Presentation:** 3 good
**Contribution:** 3 good

**Summary:**

The paper describes stochastic back propagation for image classification and object detection. They dropout certain feature maps during back propagation which helps in saving memory and speeding up training.

**Questions:**

None

**Limitations:**

Yes

**Strengths And Weaknesses:**

It is good that you have showed the SBP for a more general task of image classification and object detection, building upon the previous work on video models. I like that you have verified its application for a transformer and ConvNeXt. The analysis is exhaustive and the insights are useful. There are minor typos and grammatical mistakes. Another round of proofreading would be useful.

---

> ### Author Response · Authors · 2022-08-02
> **Response to Reviewer WMoQ**
>
> Thank you for your valuable review. We are encouraged that you find our analysis exhaustive and our insights useful.
>
> (1) “There are minor typos and grammatical mistakes.”
>
> **R**: Thank you for pointing them out. We have gone through multiple rounds of proofreading and addressed grammatical issues and typos in the revision.
>
> For example:
>
> * L8 to applying → to apply
> * L55 methods are slow down → methods slow down
> * L73 - L76 “During ... paths” → “During ... calculation”
> * L89 fully connection layer → fully connected layer
> * L97 dominants → dominates
> * L99 - L105 paragraph changed
> * L116 “SBP ... calculation” → “Unlike ... calculations”
> * L122 - L130 paragraph changed
> * L162 are no longer the exact → is no longer exact
> * L205 the higher accuracy → the higher accuracy is
> * L208 - L219 paragraph changed
> * L224 it may loss → it may lose
> * L228 - 232 “If we fix ... location” → “If we fix ... equal importance”
> * L244 activation maps includes → activation maps include
> * L307 slightly loss → slight loss
> * etc.

---

> > ### Comment · Reviewer_WMoQ · 2022-08-08
> > **Great, thanks**
> >
> > Great, thanks

---

> > > ### Author Response · Authors · 2022-08-09
> > > **Re: Great, thanks**
> > >
> > > Thank you for your time and kind words.

---

### Author Response · Authors · 2022-08-02
**General comments**

We sincerely thank all reviewers for valuable and constructive feedback. We appreciate that
* (1) *Reviewer WMoQ* finds our analysis exhaustive and our insights helpful, our work good to show the SBP for a more general task.
* (2) *Reviewer Jp2J* finds our work a welcomed contribution to NeurIPS, as our work provides a theoretical and empirical groundwork for broader usages, provides a very good argument that SBP should see more prominent usage,  demonstrates the widespread applicability to vision tasks of a reasonably straightforward concept, is well-structured and easy to understand, and the in-depth analysis of design choices provides future researchers with a strong starting point for further study.
* (3) *Reviewer yhnq* finds our paper very promising, as our work generalizes well across tasks and our method is very simple and efficient to be implemented.
* (4) *Reviewer ySTN* finds our method simple but practical, our derivations comprehensive.

We respond to all 4 reviews in separate replies.

Here we would like to mention that we have revised our paper in the updated version. Revisions are marked in blue. In summary, we have the following updates in the main paper and supplementary material:

* We expand our analysis from a simple 4-layer network to deeper networks: ViT-Tiny and ConvNeXt.
* We analyze the gradient norm and cosine similarity of model weights during the training process.
* We correct grammatical issues, typos, and errors.

---

### Meta-Review · Area_Chair_YgMX · 2022-08-27

**Recommendation:** Accept
**Confidence:** Certain

**Metareview:**

The contribution is novel and well received by the reviewers and shows that SBP should indeed be considered more often.

**Award:**

No

---

### Decision · Program_Chairs · 2022-09-14

Accept